# Acceptance of and Preference for COVID-19 Vaccination in India, the United Kingdom, Germany, Italy, and Spain: An International Cross-Sectional Study

**DOI:** 10.3390/vaccines10060832

**Published:** 2022-05-24

**Authors:** Yanqi Dong, Zonglin He, Taoran Liu, Jian Huang, Casper J. P. Zhang, Babatunde Akinwunmi, Wai-kit Ming

**Affiliations:** 1Department of Infectious Diseases and Public Health, Jockey Club College of Veterinary Medicine and Life Science, City University of Hong Kong, Hong Kong 999077, China; dongyanqi@tsinghua.edu.cn (Y.D.); t.liu.10@student.rug.nl (T.L.); 2School of Public Health, The University of Hong Kong, Hong Kong 999077, China; casperz1@connect.hku.hk; 3International School, Jinan University, Guangzhou 510632, China; hezonglin0leon@stu2015.jnu.edu.cn; 4Division of Life Science, The Hong Kong University of Science and Technology, Hong Kong 999077, China; 5Singapore Institute for Clinical Sciences (SICS), Agency for Science, Technology and Research (A*STAR), Singapore 138632, Singapore; huang_jian@sics.a-star.edu.sg; 6Department of Epidemiology and Biostatistics, School of Public Health, Faculty of Medicine, Imperial College London, London W2 1PG, UK; 7Department of Obstetrics and Gynecology Brigham and Women’s Hospital Boston, Boston, MA 02115, USA; bakinwunmi@bwh.harvard.edu; 8Center for Genomic Medicine (CGM), Massachusetts General Hospital, Harvard Medical School, Harvard University, Boston, MA 02115, USA

**Keywords:** COVID-19 vaccine, vaccine hesitancy, vaccine acceptance, COVID-19 vaccine preference

## Abstract

**Objective:** India and Europe have large populations, a large number of Coronavirus disease 2019 (COVID-19) cases, and different healthcare systems. This study aims to investigate the differences between the hesitancy toward and preference for COVID-19 vaccines in India and four European countries, namely, the United Kingdom (UK), Germany, Italy, and Spain. **Methodology:** We conducted a cross-national survey for distribution in India, the UK, Germany, Italy, and Spain. More specifically, a discrete choice experiment (DCE) was conducted to evaluate vaccine preferences, and Likert scales were used to probe the underlying factors that contribute to vaccination acceptance. Propensity score matching (PSM) was performed to directly compare India and European countries. **Results:** A total of 2565 respondents (835 from India and 1730 from the specified countries in Europe) participated in the survey. After PSM, more than 82.5% of respondents from India positively accepted the COVID-19 vaccination, whereas 79.9% of respondents from Europe had a positive attitude; however, the proportion in Europe changed to 81.6% in cases in which the vaccine was recommended by friends, family, or employers. The DCE found that the COVID-19 vaccine efficacy was the most important factor for respondents in India and the four European nations (41.8% in India and 47.77% in Europe), followed by the vaccine cost (28.06% in India and 25.88% in Europe). **Conclusion:** Although most respondents in both regions showed high acceptance of COVID-19 vaccines, either due to general acceptance or acceptance as a result of social cues, the vaccination coverage rate shows apparent distinctions. Due to the differences in COVID-19 situations, public health systems, cultural backgrounds, and vaccine availability, the strategies for COVID-19 vaccine promotion should be nation-dependent.

## 1. Introduction

Coronavirus disease 2019 (COVID-19) is now affecting more than 234 million people in 223 countries, areas, or territories as of September 2021 [1]. As of December 2021, severe acute respiratory syndrome coronavirus 2 (SARS-CoV-2), and its variants, still affect all aspects of people’s lives and pose a heavy disease burden worldwide [2].

The Delta variant, which has increased transmissibility and has caused a higher rate of severe cases [3], was first detected in India in October 2020 and has led to a massive second wave of COVID-19 infections in India [4]. Later, the World Health Organization (WHO) designated the newly detected variant termed SARS-CoV-2 Omicron (B.1.1.529) as a Variant of Concern (VOC) in November 2021, which has caused the fifth wave of pandemic around the globe, affecting more than 50% of the population in the Europe [5].

Vaccination remains one of the most cost-effective methods to prevent the development and spread of infectious diseases, and COVID-19 vaccines are considered one of the most potent means for halting the spread of the COVID-19 virus [6,7], and they are also regarded as one of the most appropriate ways for establishing herd immunity [8]. The government of India has established an administration named NEGVAC (National Expert Group on Vaccine Administration for COVID-19) to guide all aspects of COVID-19 vaccination nationally, and free vaccination against COVID-19 was also provided by the government, which urged all of its citizens to get immunized. Moreover, according to NEGVAC, the COVID-19 vaccine will first be offered to healthcare workers, frontline workers, and persons above 50 years of age, followed by persons younger than 50 years of age with associated comorbidities [9,10], which were the same strategies adopted by the selected European countries [11].

Despite the unprecedented development and rollout of COVID-19 vaccines, and the availability of various vaccines [7,12], the public’s attitude toward the COVID-19 vaccines are still unclear [13,14,15] in India and Europe. Moreover, COVID-19 vaccine hesitancy is increasing worldwide, which may influence the promotion of future vaccines [13,16]. Previous studies have shown that many people refuse or delay the COVID-19 vaccination until the vaccine’s safety is confirmed by authorities [14,17]. In addition, factors such as demographic and disease-specific characteristics also play vital roles in people’s decision-making about vaccination [16].

The recent outburst of variant-related COVID-19 infected cases, which has caused many deaths, have caused a global alert. Despite the fact that India has more than 170 million people that have been vaccinated with COVID-19 vaccines [18,19], their willingness to get the second dose, or the willingness of the unvaccinated group to get vaccinated, may still be influenced by various factors, such as distrust for the vaccine safety, unavailability of vaccinations, and cost [20,21]. Many European countries comprised the epicenter of the COVID-19 pandemic. Hence, vaccine acceptance by European residents is worth investigating to help implement a proper vaccination promotion strategy, especially when globally, the booster shots of COVID-19 vaccines have now been administered and encouraged as much as possible.

In the study, we chose Italy, Germany, Spain, and the United Kingdom, which we propose are representative of the current status quo of the Europe [22]. Italy was the first country struck by the pandemic, and it was strikingly hard-hit, with notably high cases of infection and case fatality rates. Later, following Italy, Spain was the second-most affected country in Europe by the end of March 2020. Nevertheless, as opposed to the high case-fatality rates in the two aforementioned countries, Germany, which has the largest population in Europe, except Russia, has an incredibly low fatality rate; hence it was selected. Moreover, with a similar population, the United Kingdom was selected, given its out-of-the-European-Union state. Moreover, India is one of the most hard-hit countries by the Delta variant-driven second wave of the pandemic. Although the two regions manifest distinctly different cultural and historical backgrounds, governmental administrations, healthcare systems, and COVID-19 vaccination strategies in all aspects, the two entities are being increasingly compared due to their internal diversity and international relationships [23].

The current study will pursue several objectives: (1) to conduct a comprehensive investigation into COVID-19 vaccination hesitancy in India and four selected European countries; (2) to compare the preferences for currently available COVID-19 vaccines between the two regions; and (3) to investigate the factors influencing vaccination acceptance and preference among respondents from the two regions.

## 2. Materials and Methods

### 2.1. Study Design

The present study is a cross-sectional study in which an anonymous, self-administered questionnaire (Appendix A) was disseminated via multiple online international panel providers in India, the UK, Italy, Germany, and Spain. This study was approved by the Jinan University Institutional Review Board (JNUKY-2021-004). The questionnaire was developed using Lighthouse Studio (Sawtooth Software, version 9.8.1, Sawtooth Software Inc. Provo, United States). The questionnaire was designed in English, abiding by the ISPOR Good Practice guidelines [24,25,26]. Later, the content validity and the reliability of the questions were assessed by both experts and general populations, and a group of experts were consulted to improve semantics and readability.

In the questionnaire, a total of 55 items were included, consisting of three main parts. Firstly, demographic information and social-economic information were collected, such as age, sex, education level, annual income, and occupation, followed by one set of discrete choice experiments (DCE) to investigate people’s preferences for various kinds of COVID-19 vaccines. The attributes and levels of the DCE are shown in Table 1. More specifically, the attributes included in the DCE section are vaccine types, adverse effects, efficacy, time taken for the vaccine to work, the duration of the vaccine, and the cost of vaccination. In the third part of the survey, a 7-point Likert scale was applied to evaluate people’s attitudes, acceptance, preference, knowledge, perceived risk, and benefits of the currently available vaccines. The reliability (alpha = 0.8951) and validity (Kaiser–Meyer–Olkin Measure = 0.942) of the Likert scale were assessed. This part involves seven aspects: (1) attitudes toward COVID-19 and COVID-19 vaccines; (2) perceived benefits of getting the COVID-19 vaccine; (3) perceived risks and barriers of getting the COVID-19 vaccine; (4) perceived safety and efficacy of the COVID-19 vaccine; (5) general attitude and trust for vaccines; (6) socio-economic factors; and (7) past immunization behaviors.

Before the formal collection of data, a pilot study with experts and the general population was conducted in China to assess the validity and reliability of the content of the questions, and experts were consulted to improve the readability of semantics. 

### 2.2. Respondents

The target respondents in this study were adults aged 18 and over without cognitive impairments (self-report on the first page of the survey) from India, the UK, Germany, Italy, and Spain. Through multiple online international panels, a total of 835 respondents in India and 1730 respondents in Europe were recruited, and the final sample consisted of 2565 respondents after quality control was administered. Manual checks were done to exclude incomplete and invalid questionnaires, as shown in Figure 1. 

### 2.3. Data Collection

The survey was performed from 29 January to 13 February, 2021. Upon finishing the questionnaire, the respondents would receive a random code and were asked to submit the code to the survey coordinator to verify that they were real people instead of robots. Only close-ended questions with tick boxes were provided in the questionnaire. Questions were not allowed to be skipped, and no data would be stored unless the respondents reached the final questions and submitted the random code; therefore, no missing data were inherently generated. Consent was obtained by the respondents positively responding to the question “Do you consent to participate in the following questionnaire?” and no data were obtained and stored if the respondents refused to give consent.

Educational levels were classified into four groups: (1) “low education level” meant respondents that have not finished high school education, (2) “medium” represented respondents who had finished high school, vocational school, or an equivalent degree, (3) “high” signified respondents who had completed tertiary education or had acquired a bachelor’s degree, and (4) “very high” indicated postgraduate education or above. 

When asking “How do you rate your willingness and acceptance to get vaccinated?” and “How do you rate your willingness and acceptance if your friends, family members, neighbors, and others recommend that you do so?”, respondents needed to choose from totally unwilling (0 points) to totally willing (10 points). If respondents scored greater than six on these two questions, it was regarded as general willingness and acceptance. In addition, respondents answered questions about the main sources of information about the COVID-19 vaccines. 

### 2.4. Statistical Analysis

An international cross-sectional survey was conducted to acquire the data for this research. Propensity scoring was calculated and matched to balance covariates for respondents in India and Europe to avoid potential confounding biases due to differences in baseline characteristics. PSM is a statistical technique for reducing selection bias in quasi-experimental and observational studies, which can help improve causal claims [27]. The confounding variables in this research include age, gender, educational level, and annual income.

Using central tendency (mean, median) and dispersion (standard deviation, interquartile interval) measures, descriptive statistics were used to describe the characteristics of socioeconomic status and demographic information, risk perception, pandemic impact, acceptance, attitudes, and preferences for COVID-19 vaccines. The chi-square or Fischer’s exact test was employed for univariate analysis of qualitative data, whereas for quantitative variables, the Student’s *t*-test or Mann–Whitney test was used. For qualitative data, absolute and relative frequencies were reported, whereas quantitative values were presented as the mean (standard deviation (SD)). Then, using odds ratios, standard errors, and 95% confidence intervals, multivariate logistic regression was used to find the influencing factors of vaccination acceptance (immediate or delayed acceptance) between the vaccine demand and vaccine delay groups (OR, SE, and CI, respectively). STATA, version 14.0, was used to analyze the data (Stata Corp, College Station, TX, USA). We used a conditional logit model (CLOGIT) to quantify respondents’ preferences for vaccine attributes and levels in a trade-off in general, and a further examination of participants’ preference heterogeneity across countries and regions for the DCE component was conducted [24]. The DCE was analyzed using Sawtooth Lighthouse Studio version 9.9.1 (Sawtooth Software Inc., Provo, UT, USA).

### 2.5. Scenario Analysis and Simulation

In addition, we conducted a scenario analysis and product simulation to further investigate vaccinations in terms of which features contributed the most to respondents’ preferences, and which had the greatest probability of being accepted. The base scenario was based on the vaccine variety, with every attribute’s level set to the lowest value (except the cost attribute), whereas the optimal scenario was based on the vaccine variety, with every attribute’s level set to the greatest value (except the cost attribute). The other scenarios created were based on the most up-to-date vaccine information from several clinical trials [28,29,30]. We also utilized the share of preferences as our simulation model, since this model could better predict the level of preference that any vaccine might achieve. The simulation was completed in two stages: (1) subject the respondent’s total value for the product to an exponential transformation, specifically as s=exp(utility), and (2) rescale the results to a total of 100%.

## 3. Results

### 3.1. Respondents’ Characteristics

The present study involved a large-scale self-administered online survey in India, the UK, Germany, Italy, and Spain. A total of 2565 respondents (835 from India and 1730 from Europe) have completed the survey. For the two respondent groups, most respondents were male (62.3% for India and 69.9% for Europe); around 65% of respondents from India and 95% of respondents from Europe held a bachelor’s degree or higher. More than 80% of respondents from both regions were younger than 40 years old. Most respondents had an annual salary level ranging from USD 10,000 to USD 30,000–40,000—78% in India and 75.4% in Europe. After propensity score matching for age, sex, education, and annual income, no statistically significant discrepancies could be found in the two groups of respondents from the four European countries and India in terms of baseline characteristics (*p* = 1.00 for age, sex, education, annual income, and occupation). 

### 3.2. General Hesitancy and Participants’ Vaccination History

As listed in Table 2, the pre-PSM results showed that respondents from India had a relatively lower hesitancy toward receiving COVID-19 vaccines (7.8/10) than those from European countries (7.4/10) when asked “How do you rate your willingness and acceptance to get the COVID-19 vaccination if the vaccines are generally available” (*general acceptance*). After PSM, over 82.5% of respondents from India positively accepted the COVID-19 vaccination, whereas 79.9% of respondents from Europe positively did so. Nevertheless, when recommended to get the vaccination by friends, family members or employers, and others *(acceptance under social cues*), the percentage changed to 81.6% in Europe. Most respondents in both regions claimed that they had not been infected with COVID-19 before (83.7% in India and 66.9% in Europe), and most of their friends, families, or neighborhoods had never been infected (79% in India and 73.6% in Europe).

Around 24.6% of respondents from India and 54.2% from Europe had delayed or canceled their vaccination for reasons other than illness or allergy. Moreover, 24.6% of Indian respondents and 52.4% of European respondents answered “yes” when asked whether they would accept a COVID-19 vaccine for reasons other than illness or allergy.

In total, 46.6% of Indian respondents received a recommendation from a doctor to get a COVID-19 vaccination, whereas more than half of European respondents (62%) received a recommendation from a doctor. More than 60% of Indian and European respondents received a recommendation to get vaccinated, either by the local health board and/or by friends or families.

### 3.3. Post-PSM Vaccine Preference, Attributes, and Level Importance

Figure 2 compares the relative importance of attributes in India and Europe. After PSM, it was found that respondents from both India and Europe placed the greatest emphasis on the efficacy of COVID-19 vaccines, with the latter being more evident (41.8% and 47.77% in India and Europe, respectively), followed by the cost of the vaccination (28.06% and 25.88% in India and Europe, respectively). In addition, respondents from India were also concerned about vaccine varieties, which occupied 10.06% of attribute importance and ranked as the third most important factor. In contrast, respondents from Europe were less concerned about the varieties of vaccine, which only occupied 1.24% in terms of importance. Instead, Europeans were more concerned about vaccine duration, the factor that ranked as the third most important one, accounting for 14.93%. Similarly, the start time for vaccination and adverse vaccine effects showed relatively low importance in both regions. 

Interestingly, in post-PSM results, respondents from India preferred the adenovirus vector COVID-19 vaccine, whereas respondents from Europe preferred the mRNA COVID-19 vaccine (adenovirus vector vaccines versus mRNA: OR = 0.967; 95% CI (0.924–1.011) or inactivated vaccines versus mRNA: OR = 0.955, 95% CI (0.913–0.998), as shown in Table 3. In both countries, respondents’ preferences increased with efficacy and reached a peak at 95% efficacy (versus 55%). The reduction in vaccine preference only appeared in respondents from India if the time for the vaccines to start to work was longer than 20 days in contrast to 5 days (OR = 0.875, 95% CI (0.810–0.944)). Moreover, respondents from both countries preferred longer vaccine protection times and a low level of vaccination costs.

### 3.4. Post-PSM Scenario Analysis and Uptake Likelihood Prediction

The simulation results show the simulated share of preferences for nine alternative scenarios based on real-world data from several COVID-19 vaccinations reported in large-scale clinical trials, as shown in Table 4.

The base scenario vaccine is the vaccination with the lowest preference, and 3.5% of Indian respondents were willing to accept it, whereas just 0.7% of European respondents favored it; however, we discovered that respondents from India were more inclined to favor adenovirus vector vaccines (share of choice 16.1% for Scenario 5) over mRNA vaccines (part of preference 16.1% for Scenario 5) (share of preference of 13.5% and 13.3% for Scenarios 2 and 3, respectively). The mRNA vaccination received a higher preference from European respondents (Scenario 2, share of preference of 8.1%, and Scenario 3, share of preference of 7.1%), whereas the adenovirus vector vaccines received a lower preference from European respondents.

In Scenario 8, of the share of preference study, 30.9% and 70.2% of respondents from India and Europe, respectively, would choose the hypothetical vaccination if all vaccine qualities were set to the best levels (Table 4).

### 3.5. Behavioral and Psychological Results

The results of the Likert Scale revealed that respondents had a generally positive attitude toward the benefits of COVID-19 vaccines (India: 16.7/21; Europe: 16.1/21) and were not concerned about the risks and barriers of COVID-19 vaccination (India: 10.4/21; Europe: 14.1/21); however, respondents from Europe were slightly more concerned about the vaccine’s side effects and safety than those from India (India: 10.4/21; Europe: 14.1/21).

In addition, respondents generally believed in the importance and efficacy of vaccination in disease prevention, as evidenced by high ratings for the item “In general, vaccination is effective in preventing diseases” (India: 5.8/7.0; Europe: 5.5/7.0) and 6.0/7.0 and 5.8/7.0 for respondents from India and the four European countries, respectively, for the item “In general, prevention is better than cure.”

Religion and culture were the least important socio-cultural factors in deciding whether or not to accept the COVID-19 vaccination (India: 3.1/7.0; Europe: 4.8/7.0). The item “I believe that people are endangering their health or the health of society if they do not take a COVID-19 vaccine” earned the lowest score from respondents (India 2.3/7.0; Europe: 4.2/7.0).

## 4. Discussion

The majority of the respondents from Europe and India expressed low hesitancy for COVID-19 vaccines, either in general or in response to recommendations from friends, relatives, or employers. The adenovirus vector vaccines were favored by Indian respondents, whereas European respondents favored the mRNA COVID-19 vaccines. The efficacy of COVID-19 vaccinations was considered most important by respondents from both regions, followed by the cost of vaccination.

### 4.1. The COVID-19 Vaccination Situation in India and Europe

Our study found that respondents from Europe had higher hesitancy toward COVID-19 vaccines than those from India. According to a previous survey across 15 countries, a wide range of vaccine acceptance in India was reported to be as high as 87%, ranking India the highest among all surveyed countries, whereas European respondents had a relatively lower acceptance of the COVID-19 vaccine, as low as 54% in France [31].

As of September 2021, more than 33 million confirmed infected cases and about 448,000 deaths had been reported in India [32]; however, only 51% of Indian people have been injected with at least one dose of COVID-19 vaccines, and only 20% were fully vaccinated [33]. It is estimated that herd immunity could be achieved if 70% and 85% of the population have antibodies against COVID-19 [8]. Nevertheless, the current situation of COVID-19 vaccination in India renders it hard to achieve herd immunity through active immunization due to the availability of the COVID-19 vaccines.

The Prime Minister of India called to vaccinate the entire adult population of India, and with a limited amount of vaccines supplied, a widespread shortage was incurred across the country [34]. India is regarded as a vaccine manufacturing hub worldwide, and the Serum Institute of India (SII) is the biggest vaccine manufacturing industry globally, contributing to 60% of the global vaccine supply [35,36]. Nevertheless, hundreds of thousands of Indians still only received the first dose and could not obtain a second one [37,38]. Moreover, in April 2021, the Biden administration restricted the export of raw materials needed for vaccine production, which threatened to slow India’s vaccination drive [39]. Vaccine inequality is increasing constantly, where high-income countries have around 70 times more doses per inhabitant than low-income countries [40].

Globally, the COVID-19 pandemic is still severe; therefore, strengthening cooperation between countries may not only help limit the COVID-19 pandemic but also improve vaccine development and disease treatments [41]. We found that respondents from India chose adenovirus vector vaccines, whereas European respondents favored mRNA vaccines. This conclusion could be connected to the types of vaccines that are actually available in India and Europe. The Serum Institute of India (SII), as mentioned before, provided more than half of Covishield for India, which is a kind of adenovirus vector vaccine. On the contrary, in European countries, BioNTech and Pfizer were given conditional marketing authorizations by the Commission for the vaccines developed, which produced mRNA vaccines with a high vaccine manufacturing capacity [42].

### 4.2. The Preference for Different COVID-19 Vaccines

After propensity score matching, our findings demonstrate significant disparities between respondents from India and Europe in terms of vaccination safety and efficacy, as well as perceived risks and barriers to infection. Although many kinds of COVID-19 vaccines have now been supported or administered by the WHO, vaccination programs still face many obstacles. One of the most frequently reported obstacles to vaccine acceptance is vaccine safety and efficacy [43]. Based on our results, despite high acceptance of the vaccines, the respondents from Europe are more concerned about the efficacy of the vaccine, and they may be afraid that the vaccines may not successfully prevent COVID-19.

Moreover, the perceived risks and barriers may form another obstacle, and the respondents from Europe were more concerned about the side effects, the storage, and transportation of the vaccine than those from India. This difference, when combined with their high susceptibility risks, may have led to them placing a higher value on the attribute “efficacy” and prompted a greater inclination to choose “very mild” when asked about the side effects of vaccines in the discrete choice experiment.

### 4.3. Ensuring Fast Vaccination Is the Key

Although respondents from both regions emphasized the impact of cost on their preferences, both governments have implemented policies to ensure the promotion of vaccines by providing them free of charge. As a result, the effect of the cost of COVID-19 vaccines can be ignored to some extent when promoting vaccination acceptance. Until now, the fully vaccinated rate in the four selected European countries is near 60%, and the rate of at least one dose of COVID-19 vaccine is about 70% [18]. Conducting the COVID-19 vaccination program in India is still critical and not promising [18]. High vaccination rates are essential for achieving individual and herd immunity [44]. Although each government encourages citizens to be vaccinated, the vaccination growth rate seems not to be high enough to achieve herd immunity in a brief period of time. In this sense, the vaccine allocation strategy may be the key for further increasing the vaccination rate. Age-structured mathematical models have been established to explore the efficacy of vaccine distribution [45,46,47]. For instance, Roy et al. estimated that giving those older than 60 years old their vaccination first would result in an optimized decrease in deaths [48].

### 4.4. Reducing Vaccine Hesitancy

Vaccine hesitancy is caused by many factors, including understanding the value of a vaccine, its safety and effectiveness, accessible immunization services, and lack of trust in healthcare agencies [49]. Our findings suggested that education level and public trust in government are key factors associated with acceptance of COVID-19 vaccination, whereas availability, safety, efficacy, and duration of vaccine effectiveness of different vaccines are key factors in people’s vaccine choice preferences, which showed the same result as previous studies [50,51,52,53,54]. Moreover, the spread of COVID-19-related misinformation has been reported to significantly affect vaccine hesitancy [55,56]. Previous studies have shown that the most effective intervention to deal with vaccine hesitancy is to improve people’s knowledge and awareness or attitudes toward vaccines, particularly embedding that knowledge into hospital procedures [57].

In addition, the Likert Scale score gap between the following two items, “I trust the information I receive about COVID-19 vaccines” and “I trust that my government is making decisions in my best interest with respect to what COVID-19 vaccines are provided” was found to be different. The European respondents had a higher score on these two items than those from India. Previous research has found that public trust in the government is critical for successfully implementing social policies that rely on public behavior reactions. In terms of controlling COVID-19, higher trust in the government is significantly related to the higher adoption of health behaviors, including handwashing, social distancing, and self-isolation [58,59]. 

Moreover, interventions, such as increasing access to vaccination, mandating vaccinations, and targeting healthcare workers and high-risk populations have also been proven as useful methods for solving vaccine hesitancy. On the contrary, passive intervention (such as posters and websites) and quality improvement of data collection and monitoring show the least effect on improving vaccine acceptance [57]; however, due to the complicated factors associated with vaccine hesitancy, no specific intervention strategy that can solve all kinds of vaccine hesitancy exists. It is essential to improve the understanding of the root causes of vaccine hesitancy within a country or a population subgroup by tailoring evidence-based strategies to address the root causes [60]. To promote high vaccination coverage and reduce people’s vaccine hesitancy, the Strategic Advisory Group of Experts Working Group (SAGE WG) on Vaccine Hesitancy has put forward some recommendations [61]. First, it is necessary to increase the understanding of vaccine hesitancy and its determinants. Second, the organization for accessing vaccines should be well-structured at the local, national, and global levels. Third, experience and practice sharing between different countries are also helpful for implanting new approaches to vaccine hesitancy [61].

### 4.5. Ensuring High Efficiency and Safety

In our study, we created a hypothetical vaccination that had all of the vaccine’s characteristics and was set to the best levels, after which 30.9% of Indian respondents and 70.2% of European respondents, respectively, said they would select this hypothetical vaccine. People appear to prefer vaccines with high efficiency, few side effects, high safety, and a lengthy duration, a conclusion that should motivate vaccine manufacturers and researchers to develop more effective COVID-19 vaccines.

Currently, SARS-CoV-2 and its variants are still spreading rapidly worldwide. Due to the lack of sufficient cooperation between countries and non-uniform regulations, many regions have failed to effectively control this infectious disease; therefore, enhancing international cooperation and collaboration, and avoiding vaccine nationalism, are vital for controlling the global COVID-19 pandemic. More specifically, the first measure is to strengthen cooperation between different countries to ensure a continuous supply of raw materials for vaccine production. Moreover, ready-to-use vaccines should also have increased distribution around the globe to help contain the disease in the middle- and low-income countries (LMICs). In addition, countries must cooperate with each other to ensure that the early warning system can quickly detect and contain other infectious disease outbreaks in the future [41].

## 5. Strengths and Limitations

The present study is the first one that provides a comprehensive analysis according to the vaccine hesitancy phenomenon in India and selected European countries. In addition, socio-cultural and mental factors were considered and investigated. These results may provide abundant information for policymakers. In the study, we used a close-end self-administered mode of questionnaire to prevent missed data, and we used the online panel platform, MTurk, to prevent selection bias, as previous studies have proven the census-level quality of survey data collected via MTurk [62,63]. Despite the high reliability of the data collected with MTurk, the results should still be treated with caution. 

However, this study still has limitations. First, the impartiality, the ethnic factors and current living locations of respondents were not collected in our study; hence the racial classifications and location classifications (i.e., rural vs. urban areas) of the respondents were not reported in the present study. This may lead to inadequate exploration of the socioeconomic roots underlying the acceptance and preferences of the respondents. Moreover, there may be selection bias in the present study. The language setting of the questionnaire was English, which may exclude the non-English speakers, and thus may overestimate the educational level. Hence the vaccine hesitancy shown here may not reflect the real situation; however, considering the distinct difference between the socioeconomic status of the two regions is inapplicable, so the use of PSM made possible the direct comparisons between the respondents from India and Europe, and minimized the impact of selection bias on the conclusion of our study.

## 6. Conclusions

Respondents from both India and Europe show high acceptance of the COVID-19 vaccine. The adenovirus vector vaccines were favored by Indian respondents, whereas their European counterparts favored the mRNA COVID-19 vaccines. The efficacy of COVID-19 vaccinations was considered most important by both regions, followed by the cost of vaccination. The influencing factors of vaccine hesitancy demonstrated great variability in respondents from the two regions. Multidisciplinary cooperation and global collaboration should be established to decrease vaccine hesitancy and increase vaccination coverage, and a nation-specific vaccine distribution strategy should be developed.

## Figures and Tables

**Figure 1 vaccines-10-00832-f001:**
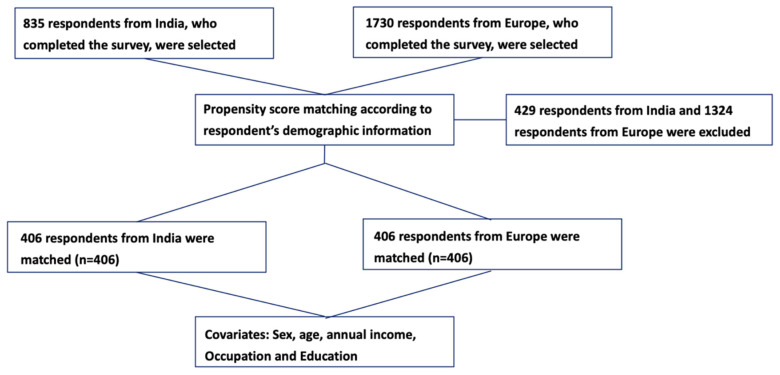
The Flow Chart of Propensity Score Matching.

**Figure 2 vaccines-10-00832-f002:**
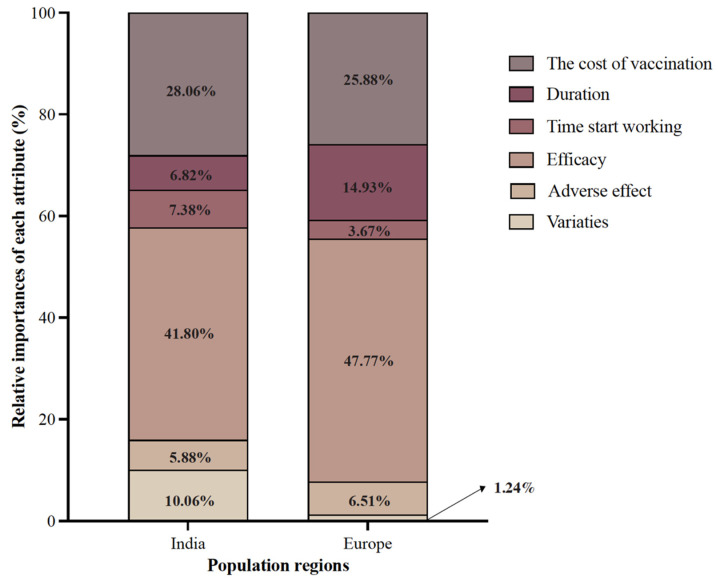
The Relative Importance of COVID-19 Vaccine Attributes: Comparison between Respondents from India and European Countries.

**Table 1 vaccines-10-00832-t001:** The attributes and levels of vaccine acceptance and preference.

Attributes	Attributes Description	Levels
Vaccine varieties	Different varieties of vaccines developed in different countries	mRNA; adenovirus vector vaccines; inactivated vaccine
Adverse effects	The extent of the adverse effect after getting vaccinated	very mild; mild; moderate
Efficacy	The efficacy of vaccines that protect the vaccinators from getting infected with COVID-19	55%; 65%; 75%; 85%; 95%
Time taken for the vaccine to start working	Time taken for the vaccine to start working after getting vaccinated	5 days; 10 days; 15 days; 20 days
The duration of time that the vaccine works	Time taken from when the vaccines start working to their invalidation	5 months; 10 months; 15 months; 20 months
The cost of vaccination	The cost of the whole vaccination process	USD 0; USD 50; USD 100; USD 150; USD 200

**Table 2 vaccines-10-00832-t002:** Participants’ demographic information, the major source of information, and acceptance.

	Unmatched			Matched ^1^		
	Europe (N = 835)	India (N = 1730)	*p*-Value	Europe (N = 406)	India (N = 406)	*p*-Value
**Sex (%)**						
Male	520 (62.3%)	1209 (69.9%)	<0.001	245 (60.3%)	240 (59.1%)	0.77
Female	314 (37.6%)	521 (30.1%)		161 (39.7%)	166 (40.9%)	
Other	1 (0.1%)	0 (0.0%)				
**Age groups (%), years**						
18–25	213 (25.5%)	352 (20.3%)	<0.001	73 (18.0%)	78 (19.2%)	0.46
26–30	181 (21.7%)	614 (35.5%)		130 (32.0%)	128 (31.5%)	
31–35	178 (21.3%)	427 (24.7%)		106 (26.1%)	98 (24.1%)	
36–40	109 (13.1%)	157 (9.1%)		53 (13.1%)	48 (11.8%)	
41–45	77 (9.2%)	86 (5.0%)		33 (8.1%)	31 (7.6%)	
46–50	41 (4.9%)	46 (2.7%)		6 (1.5%)	7 (1.7%)	
51–55	14 (1.7%)	23 (1.3%)		4 (1.0%)	8 (2.0%)	
56–60	13 (1.6%)	11 (0.6%)		1 (0.2%)	7 (1.7%)	
Above 60	9 (1.1%)	14 (0.8%)		0 (0.0%)	1 (0.2%)	
**Highest educational level (%)**						
Pre-primary education or primary school education	4 (0.5%)	2 (0.1%)	<0.001	16 (3.9%)	16 (3.9%)	0.58
Middle school education	13 (1.6%)	8 (0.5%)		14 (3.4%)	9 (2.2%)	
High school education	216 (25.9%)	34 (2.0%)		233 (57.4%)	225 (55.4%)	
Vocational school education	58 (6.9%)	47 (2.7%)		142 (35.0%)	156 (38.4%)	
Bachelor’s degree	328 (39.3%)	1105 (63.9%)		1 (0.2%)	0 (0.0%)	
Master’s degree	201 (24.1%)	529 (30.6%)				
PhD degree	15 (1.8%)	5 (0.3%)				
**Occupation (%)**						
Students	197 (23.6%)	48 (2.8%)	<0.001	70 (17.2%)	16 (3.9%)	<0.001
Managers	89 (10.7%)	360 (20.8%)		32 (7.9%)	97 (23.9%)	
Professionals	240 (28.7%)	549 (31.7%)		148 (36.5%)	115 (28.3%)	
Technicians and associate professionals	83 (9.9%)	348 (20.1%)		45 (11.1%)	83 (20.4%)	
Clerical support workers	30 (3.6%)	94 (5.4%)		12 (3.0%)	16 (3.9%)	
Service and sales workers	82 (9.8%)	159 (9.2%)		48 (11.8%)	36 (8.9%)	
Skilled agricultural, forestry and fishery workers	8 (1.0%)	41 (2.4%)		2 (0.5%)	14 (3.4%)	
Craft and related trade workers	14 (1.7%)	17 (1.0%)		7 (1.7%)	2 (0.5%)	
Plant and machine operators and assemblers	8 (1.0%)	11 (0.6%)		5 (1.2%)	2 (0.5%)	
Elementary occupations	17 (2.0%)	19 (1.1%)		10 (2.5%)	4 (1.0%)	
Armed forces occupations	7 (0.8%)	1 (0.1%)		2 (0.5%)	0 (0.0%)	
Other	60 (7.2%)	83 (4.8%)		25 (6.2%)	21 (5.2%)	
**Annual income (%)**						
Under USD 10,000	173 (22.4%)	533 (31.1%)	<0.001	100 (24.6%)	100 (24.6%)	0.99
USD 10,001–20,000	173 (22.4%)	343 (20.0%)		101 (24.9%)	96 (23.6%)	
USD 20,001–30,000	136 (17.6%)	254 (14.8%)		64 (15.8%)	64 (15.8%)	
USD 30,001–40,000	121 (15.6%)	162 (9.5%)		67 (16.5%)	67 (16.5%)	
USD 40,001–50,000	72 (9.3%)	147 (8.6%)		40 (9.9%)	40 (9.9%)	
USD 50,001–60,000	39 (5.0%)	133 (7.8%)		19 (4.7%)	19 (4.7%)	
USD 60,001–70,000	19 (2.5%)	114 (6.7%)		9 (2.2%)	14 (3.4%)	
Above USD 70,000	41 (5.3%)	26 (1.5%)		6 (1.5%)	6 (1.5%)	
**Acceptance of vaccination ^2^, mean (SD)**	7.8 (2.9)	7.4 (2.6)	0.001	7.7 (2.9)	7.4 (2.5)	0.054
**Acceptance of vaccination under social cues ^2^, mean (SD)**	7.7 (2.8)	7.5 (2.5)	0.14	7.6 (2.8)	7.4 (2.4)	0.27
**Self ever infected with COVID-19 (%)**						
Yes	131 (15.7%)	556 (32.1%)	<0.001	68 (16.7%)	131 (32.3%)	<0.001
No	699 (83.7%)	1157 (66.9%)		336 (82.8%)	271 (66.7%)	
Not to answer	5 (0.6%)	17 (1.0%)		2 (0.5%)	4 (1.0%)	
**Friend family or community ever infected (%)**						
Yes	660 (79.0%)	1274 (73.6%)	0.005	327 (80.5%)	317 (78.1%)	0.68
No	168 (20.1%)	447 (25.8%)		76 (18.7%)	86 (21.2%)	
Not to answer	7 (0.8%)	9 (0.5%)		3 (0.7%)	3 (0.7%)	
**Marital status (%)**						
Single	380 (45.5%)	550 (31.8%)	<0.001	177 (43.6%)	118 (29.1%)	<0.001
Married	334 (40.0%)	1165 (67.3%)		170 (41.9%)	283 (69.7%)	
Divorced	20 (2.4%)	8 (0.5%)		9 (2.2%)	2 (0.5%)	
Other	95 (11.4%)	2 (0.1%)		47 (11.6%)	1 (0.2%)	
Not to answer	6 (0.7%)	5 (0.3%)		3 (0.7%)	2 (0.5%)	
**Source of information for COVID-19 vaccines (%)**						
Healthcare provider	347 (41.6%)	790 (45.7%)	0.050	178 (43.8%)	185 (45.6%)	0.62
CDC or public health department	247 (29.6%)	466 (26.9%)	0.16	105 (25.9%)	114 (28.1%)	0.48
News reports	622 (74.5%)	1127 (65.1%)	<0.001	295 (72.7%)	266 (65.5%)	0.028
Social media	364 (43.6%)	1141 (66.0%)	<0.001	176 (43.3%)	256 (63.1%)	<0.001
Friends or family members	280 (33.5%)	856 (49.5%)	<0.001	143 (35.2%)	184 (45.3%)	0.003
Employers	59 (7.1%)	285 (16.5%)	<0.001	33 (8.1%)	55 (13.5%)	0.013
Pharmaceutical company advertisement	25 (3.0%)	120 (6.9%)	<0.001	7 (1.7%)	27 (6.7%)	<0.001
Other	25 (3.0%)	3 (0.2%)	<0.001	9 (2.2%)	0 (0.0%)	0.003

^1^ Matching was done by calculating the propensity score for confounding variables including sex, age, educational level, occupation, and annual income. ^2^ Likert scorings, ranging from 0 to 10, with 0 being totally unwilling and 10 being totally willing. SD, standard deviation; CDC, Centers for Disease Control and Prevention; COVID-19, Coronavirus disease 2019.

**Table 3 vaccines-10-00832-t003:** Comparison between India and Europe in terms of Attribute Level Utility and Odds Ratios.

Comparison between India and Europe in Attributes Levels Utility and Odds Ratios											
	India (N = 406)						Europe (N = 406)				
Attributes	Variable	Coefficient	Std. Error	OR	95% CI	*p*-Value	Coefficient	Std. Error	OR	95% CI	*p*-Value
**Varieties**	mRNA	Reference ^1^									
	Adenovirus vector vaccines	0.080	0.028	1.058	1.000–1.118	0.005	−0.007	0.023	0.967	0.924–1.011	0.756
	Inactivated vaccine	−0.103	0.029	0.881	0.833–0.932	0.000	−0.020	0.023	0.955	0.913–0.998	0.390
**Adverse effect**	very mild	Reference									
	mild	0.055	0.029	1.061	1.002–1.122	0.056	0.052	0.023	0.957	0.915–1.002	0.024
	moderate	−0.052	0.029	0.953	0.901–1.008	0.073	−0.15	0.023	0.783	0.748–0.819	0.000
**Efficacy**	55%	Reference									
	65%	−0.211	0.041	1.142	1.055–1.237	0.000	−0.55	0.034	1.407	1.317–1.503	0.000
	75%	−0.042	0.046	1.353	1.237–1.480	0.363	0.075	0.035	2.622	2.449–2.807	0.032
	85%	0.181	0.040	1.690	1.562–1.827	0.000	0.456	0.031	3.837	3.612–4.077	0.000
	95%	0.415	0.044	2.137	1.961–2.328	0.000	0.906	0.034	6.023	5.632–6.443	0.000
**Time for the** **vaccine to start working**	5 days	Reference									
	10 days	0.020	0.035	0.958	0.894–1.027	0.578	−0.071	0.028	0.871	0.824–0.921	0.013
	15 days	−0.011	0.035	0.929	0.868–0.996	0.761	0.022	0.028	0.956	0.906–1.009	0.428
	20 days	−0.071	0.039	0.875	0.810–0.944	0.067	−0.018	0.031	0.919	0.865–0.976	0.563
**The duration of vaccine effectiveness**	5 months	Reference									
	10 months	−0.075	0.036	0.935	0.872–1.003	0.037	−0.089	0.028	1.275	1.206–1.348	0.002
	15 months	0.049	0.037	1.059	0.984–1.139	0.185	0.193	0.029	1.692	1.598–1.791	0.000
	20 months	0.033	0.036	1.041	0.970–1.118	0.362	0.229	0.028	1.753	1.658–1.852	0.000
**The cost of** **vaccination**	USD 0	Reference									
	USD 50	0.150	0.041	0.892	0.824–0.966	0.000	0.192	0.032	0.674	0.633–0.717	0.000
	USD 100	−0.040	0.039	0.738	0.684–0.796	0.308	−0.147	0.031	0.480	0.452–0.510	0.000
	USD 150	−0.130	0.043	0.674	0.619–0.734	0.003	−0.246	0.034	0.435	0.406–0.465	0.000
	USD 200	−0.245	0.046	0.601	0.549–0.658	0.000	−0.386	0.037	0.378	0.351–0.407	0.000

^1^ Reference: The variable is a reference variable.

**Table 4 vaccines-10-00832-t004:** Share of preference and scenario analysis results.

Share of Preference	Base Scenario	Scenario 1	Scenario 2	Scenario 3	Scenario 4	Scenario 5	Scenario 6	Scenario 7	Scenario 8
**INDIA**									
**1.** **Vaccine varieties**	Inactivated vaccine	Adenovirus vector vaccines	mRNA	mRNA	Inactivated vaccine	Adenovirus vector vaccines	Adenovirus vector vaccines	Adenovirus vector vaccines	mRNA
**2.** **Adverse effect**	moderate	very mild	moderate	mild	very mild	mild	very mild	very mild	very mild
**3.** **Efficacy**	55%	75%	95%	95%	75%	95%	65%	75%	95%
**4.** **Time taken for the vaccine to start working**	20 days	20 days	20 days	20 days	10 days	20 days	5 days	5 days	5 days
**5.** **The duration of vaccine effectiveness**	5 months	5 months	5 months	5 months	5 months	5 months	5 months	5 months	20 months
**6.** **The cost of vaccination**	USD 50	USD 50	USD 50	USD 50	USD 50	USD 50	USD 50	USD 50	USD 50
**7.** **Share of preference**	3.5%	4.0%	13.5%	13.3%	3.8%	16.1%	7.7%	7.3%	30.9%
**EUROPE**									
**1.** **Vaccine varieties**	Adenovirus vector vaccines	Adenovirus vector vaccines	mRNA	mRNA	Inactivated vaccine	Adenovirus vector vaccines	Adenovirus vector vaccines	Adenovirus vector vaccines	Adenovirus vector vaccines
**2.** **Adverse effect**	moderate	very mild	moderate	mild	very mild	mild	very mild	very mild	very mild
**3.** **Efficacy**	55%	75%	95%	95%	75%	95%	65%	75%	95%
**4.** **Time for the vaccine to start working**	20 days	20 days	20 days	20 days	10 days	20 days	5 days	5 days	5 days
**5.** **The duration of vaccine effectiveness**	5 months	5 months	5 months	5 months	5 months	5 months	5 months	5 months	20 months
**6.** **The cost of vaccination**	USD 50	USD 50	USD 50	USD 50	USD 50	USD 50	USD 50	USD 50	USD 50
**7.** **Share of preference**	0.7%	1.5%	8.1%	7.1%	3.1%	5.1%	1.8%	2.5%	70.2%

## Data Availability

Data are available upon reasonable request by emailing the corresponding author.

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
