# Peer review of "Acceptance of and Preference for COVID-19 Vaccination in India, the United Kingdom, Germany, Italy, and Spain: An International Cross-Sectional Study"

_vaccines, 2022, doi:10.3390/vaccines10060832_

Round 1
Reviewer 1 Report
The manuscript reports results of a survey carried out in different countries including several European and India, as well as the descriptive analysis of these results.
Such kind of information is definitely useful a reference material for the organization of public health protection measures that is not limited by the case of COVID-19 but applicable for fighting present and future diseases.
As a special point, I would like to highlight the discussion of sources of information on vaccination in the comparison to the governmental administrating. This information gives some hints on the public reaction on organization of the public health protection measures and their efficiency.
Another strong point is a possibility to compare the social reaction in countries with different economical situations, heath protection systems, and cultural specificity.
Thus, I recommend accepting this manuscript.
Author Response
Response to comment 1:
We thank the reviewer for the high compliment on our manuscript! Thank you.
Reviewer 2 Report
MS: Acceptance of and preference for COVID-19 vaccination in India and Four Selected European countries: an international 3 cross-sectional study
Special issue: Vaccination Strategies as We Face another Year of the COVID-19 Pandemic
Minor changes
The paper represents acceptance y preference for COVID-19 vaccination in India, the UK, Germany, Italy, and Spain.
Please authors attend the following
- Why did you choose these countries?
- How was validated the pilot assay? What language?
- Which vaccine strategies were used in India vs some countries of Europe
Data could be presented in results part in a friendlier manner
There are works published that must be cited and compared in the discussion session, authors could make comparisons and to improve discussion and conclusions
Author Response
Minor comment 1:
The paper represents acceptance and preference for COVID-19 vaccination in India, the UK, Germany, Italy, and Spain.
Please authors attend the following
Why did you choose these countries?
| Response to minor comment 1:
Thank you, these countries are representative during the pandemic, and we believed that the comparison between these countries may help us better understand the different attitudes and preferences for COVID-19 vaccines in respondents from the two regions. We have revised the manuscript accordingly on Line 95-108, as attached below:
(Line 95-108):
In the study, we chose Italy, Germany, Spain, and the United Kingdom, which we proposed that can represent the current status quo of the Europe(22). Italy was the first country struck by the pandemic, and it was hard-hit strikingly with notably high cases of infection and case fatality rates. Later, following Italy, Spain was the second-most affected country in Europe by the end of March 2020. Nevertheless, as opposed to the high case-fatality rates in the two countries, Germany, which has the most population in Europe except Russia, has an incredibly low fatality rate, hence selected. Moreover, with similar amount of population, the United Kingdom was selected considering its out-of-the-European-Union state. Moreover, India is one of the most hard-hit countries by the Delta variant-driven second wave of the pandemic. Despite that these two regions manifest with distinctly different cultural and historical backgrounds, governmental administrating, healthcare system, COVID-19 vaccination strategies from all aspects, the two entities are being increasingly compared considering their internal diversity and international relationships(23).
Minor comment 2:
How was validated the pilot assay? What language?
| Response to minor comment 2:
Thanks for the comment. The questionnaire was designed in English abiding by the ISPOR Good Practice guidelines. Later, the content validity and the reliability of the questions were assessed by both experts and general populations, and a group of experts were consulted to improve semantics and readability. We have revised the manuscript accordingly on line 120-123, as attached below:
The questionnaire was designed in English abiding by the ISPOR Good Practice guidelines(24-26). Later, the content validity and the reliability of the questions were assessed by both experts and general populations, and a group of experts were consulted to improve semantics and readability.
Minor comment 3:
Which vaccine strategies were used in India vs some countries of Europe
| Response to minor comment 3:
Thanks for the comment. We have added the following contents:
(Line 68-76)”The government of India has constituted an administration named National Expert Group on Vaccine Administration for COVID-19 (NEGVAC) to guide the COVID-19 vaccination nationally, free vaccination against COVID-19 were also provided by the government and urged all of its citizens to be immunized. Besides, according to NEGVAC, the COVID-19 vaccine will be offered first to healthcare workers, frontline workers, and to persons above 50 years of age, followed by persons younger than 50 years of age with associated comorbidities(9,10),which was the same strategies with selected European countries”(11).
Minor comment 4:
Data could be presented in results part in a friendlier manner
| Response to minor comment 4:
Thanks for the comment. We have revised the Table and figures accordingly.
Minor comment 5:
There are works published that must be cited and compared in the discussion session, authors could make comparisons and to improve discussion and conclusions
| Response to minor comment 5:
Thanks for the comments, We have added the following contents :
4.1. The COVID-19 vaccination situation in India and Europe
In our study, we found that respondents from Europe had higher hesitancy toward COVID-19 vaccines than those from India. According to a previous survey across 15 countries, a wide range of vaccine acceptance in India was reported to be as high as 87%, ranking India the highest among all surveyed countries, while European respondents had a relatively lower acceptance of the COVID-19 vaccine, as low as 54% in France (31).
(line 311-327)
“As of September 2021, more than 33 million confirmed infected cases and about 448,000 deaths have been reported in India (32); however, only 51% of Indian people have been injected with at least one dose of COVID-19 vaccines, and only 20% were fully vaccinated (33). It is estimated that herd immunity could be achieved if 70% and 85% of the population has antibodies against COVID-19 (8). Nevertheless, the current situation of COVID-19 vaccination in India renders it hard to achieve herd immunity through active immunization due to the availability of the COVID-19 vaccines.
The Prime Minister of India called to vaccinate the entire adult population of India, and with limited amount of vaccines supplied, a widespread shortage was incurred across the country (34). India is regarded as the vaccine manufacturing hub worldwide, and Serum Institute of India (SII) is the biggest vaccine manufacturing industry globally, contributing 60% of the global vaccine supply (35, 36). Nevertheless, hundreds of thousands of Indians still only received the first dose, and could not obtain a second one (37,38). Besides, in April 2021, the Biden administration restricted the export of raw materials needed for vaccine production, which threatened to slow India’s vaccination drive (39). The vaccine inequality is increasing constantly, where the high-income countries have around 70 times more doses per inhabitant than those in the low-income countries (40).”
(Line 355-368)
“Despite the fact that respondents from both regions emphasized the impact of cost on their preferences, both governments have implemented policies to ensure the promotion of vaccines by providing them free of charge. As a result, the effect of the cost of COVID-19 vaccines can be ignored to some extent when promoting vaccination acceptance. Until now, the fully vaccinated rate in the four selected European countries is near 60%, and the rate of at least one dose of COVID-19 vaccine is about 70% (44). Conducting the COVID-19 vaccination program in India is still critical and not promising (18). High vaccination rates are essential for achieving individual and herd immunity (45). Although each government encourages citizens to be vaccinated, the vaccination growth rate seems not to be high enough to achieve herd immunity in a brief period of time. In this sense, the vaccine allocation strategy may be the key for further increasing the vaccination rate. Age-structured mathematical models have been established to explore the efficacy of vaccine distribution (46-48). For instance, Roy et al., estimated that giving those older than 60 years old vaccination first would result in the optimized decrease in deaths (49).”
(Line 370-380)
“Vaccine hesitancy is caused by many factors, including understanding the value of a vaccine, its safety and effectiveness, accessible immunization services, and lack of trust in healthcare agencies, among others(50). Our findings suggested that education level, public trust in government are key factors associated with acceptance of COVID-19 vaccination, while availability, safety, efficacy and duration of vaccine effectiveness of different vaccines are key factors in people's vaccine choice preferences, which showed the same result with previous studies (51-55). Moreover, the spread of COVID-19-related misinformation has been reported to significantly affect the vaccine hesitancy (56,57). Previous studies have shown that the most effective intervention to deal with vaccine hesitancy is to improve people's knowledge and awareness or attitudes toward vaccines, particularly embedding that knowledge into hospital procedures (58).
(Line 381-389)
In addition, the Likert Scale score gap of the following two items “I trust the information I receive about COVID-19 vaccines.” and “I trust that my government is making decisions in my best interest with respect to what COVID-19 vaccines are provided” was found to be different. The European respondents had a higher score on these two items than those respondents from India. Previous research has found that public trust in the government is critical for successfully implementing social policies that rely on public behavior reactions. In COVID-19 control, higher trust in the government is significantly related to higher adoption of health behaviors, including handwashing, social distancing, and self-isolation (59,60).”
Strengths and limitations
(Line 424-444)
“The present study is the first one that provides a comprehensive analysis according to the vaccine hesitancy phenomenon in India and selected European countries. Also, socio-cultural and mental factors were considered and investigated. These results may provide abundant information for policymakers. In the study, we used a close-end self-administered mode of questionnaire to prevent missed data and used online panel platform like Mturk to prevent selection bias, as previous study has proven the census-level quality of survey data collected via Mturk (63.64). Despite the high reliability of the data collected with MTurk, the results should still be treated with caution.
However, this study still has limitations. First, considering the impartiality, the ethnic factors and current living locations of respondents were not collected in our study. Hence the racial classifications and location classifications (i.e., rural vs. urban areas) of the respondents were not reported in the present study. This may lead to inadequate exploration of the socioeconomic roots underlying the acceptance and preferences of the respondents. Also, there may be selection bias in the present study. The language setting of the questionnaire was English, which may exclude the non-English speakers, which may overestimate the educational level. Hence the vaccine hesitancy may not reflect the real situation. However, considering the distinct difference between the socioeconomic status between the two regions is inapplicable, so the use of PSM made possible the direct comparisons between the respondents from India and Europe, and minimized the impact of selection bias on the conclusion of our study.”
“Conclusion
(Line 447-454)
Respondents from both India and Europe show high acceptance of the COVID-19 vaccine. The adenovirus vector vaccines were favored by Indian respondents, while European counterparts favored the mRNA COVID-19 vaccines. The efficacy of COVID-19 vaccinations was considered most important by both regions, followed by the cost of vaccination. The influencing factors of vaccine hesitancy demonstrated great variability in respondents from the two regions. Multidisciplinary cooperation and global collaboration should be established to decrease vaccine hesitancy and increase vaccination coverage, and nation-specific vaccine distribution strategy shall be developed. ”
Reviewer 3 Report
This article presents the results of a survey on hesitancy and preference regarding coronavirus vaccines, in India and four European countries, namely the United Kingdom, Germany, Italy and Spain. The study was conducted using a digital questionnaire distributed vis multiple online international panel of providers. The questionnaire had three parts with in total 55 items: one part addressed demographic parameters, one part included a set of discrete choice experiments, and one part included various parameters regarding the individual perception of vaccine. In total, 2565 individuals were included, 835 individuals from India and 1730 from India, sampled between January 29 and February 13, 2019. Main outcomes were a similar and high positive response to vaccination, and an adenovirus-based vaccine was more preferred in India and a mRNA-vaccine in European countries. The analysis was extended by nine alternative scenarios for vaccination, which were based on actual data from vaccinations performed worldwide. Amongst others, this scenario analysis confirmed the preference for adenovirus-based vaccine by People in India, and mRNA-based vaccines by people in Europe.
This is an interesting report, which adds to many studies conducted so far on vaccine hesitancy. The evaluation of the population in India and comparison with the European population, using rather new methodology is new and attractive. The design and conduct of the study are well performed, and the data presentation and interpretation is clear.
There are a number of points that need to addressed in revision:
- It is assumed that the questionnaire and communication with the respondents was done in the English language. Considering the native language in the various countries, this creates bias, with as result a selection from the population that is enrolled. Note that the questionnaire was rather large-sized and may have been difficult to understand in-full. This is clear from Table 2, showing an over-representation of people with a high education level. This point needs to be discussed in the section on strength and weaknesses.
- Following this point it may seems logical to assume that this bias also results in a selection of people living in urban areas, and not rural areas. This also can affect hesitancy to vaccination.
- The selection of countries in Europe needs a rationale.
- It needs to be stated whether there was any information provided together with the questionnaire.
- It is advised to include the questionnaire in the manuscript as supplemental information.
- The difference in vaccine preference, adenovirus-based or mRNA based, is likely ascribed to availability of vaccines. This possible explanation is first presented relatively late in the manuscript (section 4.5). Also, it needs to be illustrated by some data on availability.
- Strength and limitations: The authors write “First, the ethnic factors of respondents were not collected and thus omitted in our study. Second, as the main collecting tool in our study, the results should still be treated with ccaution, although some studies have proven the high reliability of the data collected with MTurk”. These statements need further explanation by giving data.
- The tables are difficult to read, and need some editing. It is expected that this will be done by the processing of the manuscript.
Author Response
Minor comment 1:
This article presents the results of a survey on hesitancy and preference regarding coronavirus vaccines, in India and four European countries, namely the United Kingdom, Germany, Italy and Spain. The study was conducted using a digital questionnaire distributed vis multiple online international panel of providers. The questionnaire had three parts with in total 55 items: one part addressed demographic parameters, one part included a set of discrete choice experiments, and one part included various parameters regarding the individual perception of vaccine. In total, 2565 individuals were included, 835 individuals from India and 1730 from India, sampled between January 29 and February 13, 2019. Main outcomes were a similar and high positive response to vaccination, and an adenovirus-based vaccine was more preferred in India and a mRNA-vaccine in European countries. The analysis was extended by nine alternative scenarios for vaccination, which were based on actual data from vaccinations performed worldwide. Amongst others, this scenario analysis confirmed the preference for adenovirus-based vaccine by People in India, and mRNA-based vaccines by people in Europe.
This is an interesting report, which adds to many studies conducted so far on vaccine hesitancy. The evaluation of the population in India and comparison with the European population, using rather new methodology is new and attractive. The design and conduct of the study are well performed, and the data presentation and interpretation is clear.
There are a number of points that need to addressed in revision:
- It is assumed that the questionnaire and communication with the respondents was done in the English language. Considering the native language in the various countries, this creates bias, with as result a selection from the population that is enrolled. Note that the questionnaire was rather large-sized and may have been difficult to understand in-full. This is clear from Table 2, showing an over-representation of people with a high education level. This point needs to be discussed in the section on strength and weaknesses.
| Response to minor comment 1:
We thank the reviewer for the comments, and we agree with the reviewer. We have carefully revised it according to the comment. Specifically, we have revised both the method and limitation sections on line 120-123 and Line 438-444, as attached below:
(line 120-123):
“The questionnaire was designed in English abiding by the ISPOR Good Practice guidelines (24-26). Later, the content validity and the reliability of the questions were assessed by both experts and general populations, and a group of experts were consulted to improve semantics and readability.”
(line 438-444)
“Also, there may be selection bias in the present study. The language setting of the questionnaire was English, which may exclude the non-English speakers, which may overestimate the educational level. Hence the vaccine hesitancy may not reflect the real situation. However, considering the distinct difference between the socioeconomic status between the two regions is inapplicable, so the use of PSM made possible the direct comparisons between the respondents from India and Europe, and minimized the impact of selection bias on the conclusion of our study.”
Minor comment 2:
- Following this point it may seems logical to assume that this bias also results in a selection of people living in urban areas, and not rural areas. This also can affect hesitancy to vaccination.
| Response to minor comment 2
Thank you, we have also noticed this issue, that people living in urban and rural areas may have vaccine hesitancy to a different extent. Therefore, we have also added relative discussions in our limitation section, in lines 433-444. For convenience, the sentence added is as below:
“However, this study still has limitations. First, considering the impartiality, the ethnic factors and current living locations of respondents were not collected in our study. Hence the racial classifications and location classifications (i.e., rural vs. urban areas) of the respondents were not reported in the present study. This may lead to inadequate exploration of the socioeconomic roots underlying the acceptance and preferences of the respondents. Also, there may be selection bias in the present study. The language setting of the questionnaire was English, which may exclude the non-English speakers, which may overestimate the educational level. Hence the vaccine hesitancy may not reflect the real situation. However, considering the distinct difference between the socioeconomic status between the two regions is inapplicable, so the use of PSM made possible the direct comparisons between the respondents from India and Europe, and minimized the impact of selection bias on the conclusion of our study. ”
Minor comment 3:
- The selection of countries in Europe needs a rationale.
| Response to minor comment 3:
Thank you, and we we have added the following contents to answer why we choose these countries:
(Line 95-108:)
In the study, we chose Italy, Germany, Spain, and the United Kingdom, which we proposed that can represent the current status quo of the Europe (22). Italy was the first country struck by the pandemic, and it was hard-hit strikingly with notably high cases of infection and case fatality rates. Later, following Italy, Spain was the second-most affected country in Europe by the end of March 2020. Nevertheless, as opposed to the high case-fatality rates in the two countries, Germany, which has the most population in Europe except Russia, has an incredibly low fatality rate, hence selected. Moreover, with similar amount of population, the United Kingdom was selected considering its out-of-the-European-Union state. Moreover, India is one of the most hard-hit countries by the Delta variant-driven second wave of the pandemic. Despite that these two regions manifest with distinctly different cultural and historical backgrounds, governmental administrating, healthcare system, COVID-19 vaccination strategies from all aspects, the two entities are being increasingly compared considering their internal diversity and international relationships (23).
Minor comment 4 & 5:
- It needs to be stated whether there was any information provided together with the questionnaire.
- It is advised to include the questionnaire in the manuscript as supplemental information.
| Response to minor comment 4 & 5:
Thanks for the comment. We have included the questionnaire in the supplementary materials.
Minor comment 6:
- The difference in vaccine preference, adenovirus-based or mRNA based, is likely ascribed to availability of vaccines. This possible explanation is first presented relatively late in the manuscript (section 4.5). Also, it needs to be illustrated by some data on availability.
| Response to minor comment 6:
Thanks for the comment. We have moved this content forward in the manuscript, and we have added the following contents on Line 318-337:
(Line 318-337):
“The Prime Minister of India called to vaccinate the entire adult population of India, and with limited amount of vaccines supplied, a widespread shortage was incurred across the country (34). India is regarded as the vaccine manufacturing hub worldwide, and Serum Institute of India (SII) is the biggest vaccine manufacturing industry globally, contributing 60% of the global vaccine supply (35, 36). Nevertheless, hundreds of thousands of Indians still only received the first dose, and could not obtain a second one (37, 38). Besides, in April 2021, the Biden administration restricted the export of raw materials needed for vaccine production, which threatened to slow India’s vaccination drive (39). The vaccine inequality is increasing constantly, where the high-income countries have around 70 times more doses per inhabitant than those in the low-income countries (40).
Globally, the COVID-19 pandemic is still severe. Therefore, strengthening cooperation between countries may not only help limit the COVID-19 pandemic but also improve vaccine development and disease treatments (41). We found that respondents from India chose adenovirus vector vaccines, whereas European respondents favored mRNA vaccines. This conclusion could be connected to the types of vaccines that are actually available in India and Europe. Serum Institute of India (SII), as mentioned before, provided more than half of Covishield for India, which is a kind of adenovirus vector vaccine. On the contrary, in European countries, BioNTech and Pfizer were given conditional marketing authorizations by the Commission for the vaccines developed, which produced mRNA vaccines with high vaccine manufacturing capacity(42).”
Minor comment 7:
- Strength and limitations: The authors write “First, the ethnic factors of respondents were not collected and thus omitted in our study. Second, as the main collecting tool in our study, the results should still be treated with ccaution, although some studies have proven the high reliability of the data collected with MTurk”. These statements need further explanation by giving data.
| Response to minor comment 7:
Thanks for the comment. We have revised this part accordingly on line 427-444, as attached below:
(Line 427-444:)
In the study, we used a close-end self-administered mode of questionnaire to prevent missed data and used online panel platform like Mturk to prevent selection bias, as previous study has proven the census-level quality of survey data collected via Mturk (63, 64). Despite the high reliability of the data collected with MTurk, the results should still be treated with caution.
However, this study still has limitations. First, considering the impartiality, the ethnic factors and current living locations of respondents were not collected in our study. Hence the racial classifications and location classifications (i.e., rural vs. urban areas) of the respondents were not reported in the present study. This may lead to inadequate exploration of the socioeconomic roots underlying the acceptance and preferences of the respondents. Also, there may be selection bias in the present study. The language setting of the questionnaire was English, which may exclude the non-English speakers, which may overestimate the educational level. Hence the vaccine hesitancy may not reflect the real situation. However, considering the distinct difference between the socioeconomic status between the two regions is inapplicable, so the use of PSM made possible the direct comparisons between the respondents from India and Europe, and minimized the impact of selection bias on the conclusion of our study.
Minor comment 8:
- The tables are difficult to read, and need some editing. It is expected that this will be done by the processing of the manuscript.
| Response to minor comment 8: Thanks for the comments. The format was altered by the processing of the manuscript. We have revised the table format accordingly.